# Towards an Eco-Friendly Coffee Rust Control: Compilation of Natural Alternatives from a Nutritional and Antifungal Perspective

**DOI:** 10.3390/plants11202745

**Published:** 2022-10-17

**Authors:** Nora E. Torres Castillo, Yovanina Aguilera Acosta, Lizeth Parra-Arroyo, María Adriana Martínez-Prado, Verónica M. Rivas-Galindo, Hafiz M. N. Iqbal, A. Damiano Bonaccorso, Elda M. Melchor-Martínez, Roberto Parra-Saldívar

**Affiliations:** 1School of Engineering and Sciences, Tecnologico de Monterrey, Monterrey 64849, Mexico; 2Department of Chemical and Biochemical Engineering, Tecnologico Nacional de México-Instituto Tecnológico de Durango, Blvd. Felipe Pescador 1830 Ote, Durango 34080, Mexico; 3Departamento de Química Analítica, Facultad de Medicina, Universidad Autonoma de Nuevo Leon, Monterrey 66455, Mexico; 4Institute of Advanced Materials for Sustainable Manufacturing, Tecnologico de Monterrey, Monterrey 64849, Mexico; 5School of Chemistry, University of St Andrews, North Haugh, St Andrews KY16 9ST, UK

**Keywords:** *Hemileia vastatrix*, high mountain coffee, shaded production systems, *Coffea arabica*

## Abstract

*Hemileia vastatrix* (HV) is the pathogen responsible for the coffee leaf rust (CLR) disease that has spread globally. CLR causes losses of up to a billion dollars annually and affects all types of crops regardless of their production regime (organic or inorganic). Additionally, smallholders produce approximately 80% of coffee in developing countries. The condition causes losses of up to a billion dollars annually. It affects all types of crops regardless of their production regime (organic or inorganic). Approximately 80% of coffee is produced by smallholders in developing countries. Until the 90s, shaded-production systems and native varieties were encouraged; however, the rapid spread of CLR has forced farmers to migrate towards inorganic schemes, mainly due to a lack of knowledge about natural alternatives to pesticides that can be implemented to control HV. Therefore, the purpose of this article is to compile the currently existing options, emphasizing two key factors that guarantee efficient rust control: selective fungicidal activity against HV and the nutrition of coffee crops. Thus, by comprehending how these natural compounds (such as plant, bacteria, fungi, animals, or algae metabolites) impact coffee rust proliferation. Furthermore, since a various range of biochar effects contributes to the control of foliar fungal pathogens through modification of root exudates, soil properties, and nutrient availability, which influence the growth of antagonist microorganisms, we present a review of the pathogen-suppressive effects of biochar, and new control strategies suitable for organic schemes can be developed.

## 1. Introduction

Coffee has been a crop of socioeconomic importance over history. Currently, around 25 million smallholder farmers are responsible for 80% of the world’s coffee production, while more than 125 million people depend on coffee for their livelihoods [1]. Over the past decade, the value of the global coffee industry has almost doubled to $90 billion, and more than 2 billion cups of coffee are currently consumed worldwide daily. According to the International Coffee Organization, 169.34 million bags were produced globally from 2019–2020 [1,2,3].

Hence, the market is expected to grow at an average annual rate of 5.32% between 2020 and 2024 [3]. Furthermore, due to its positive relationship with economic growth and the gross domestic product of most producing countries, it has attracted increasing global attention [4].

Specifically, *Coffea arabica* is the most common coffee specie cultivated, and it is regarded as a strategic crop because its production employs more than 500,000 smallholders from 14 states of Mexico [5,6]. However, in environments undergoing constant transformation due to human activities, such as deforestation and urbanization, agroforest coffee farming is threatened by various challenges nowadays [7].

Various parasites and diseases have afflicted coffee crops, the most significant being rust, which reduced national coffee production [7]. The *fungus Hemileia vastatrix* (HV) causes coffee leaf rust (CLR), a biotrophic pathogen (phytoparasite) that affects mainly the leaves of the specie *Coffea arabica* and is considered the most critical disease in cultivation worldwide. In summary, CLR alone has caused profit losses totaling over $3 billion and caused nearly 2 million farmers to abandon their property from 2012 to 2017 worldwide [8].

This fungus causes defoliation, reducing the coffee trees’ photosynthetic capacity and yield. The severity of the disease is affected by factors such as climate (including the effect of altitude), shade, soil fertility, and canopy architecture [9,10]. The economic impact of HV is not only due to a reduction in quantity and quality of production but also a consequence of undertaking costly management measures in sensitive cultivars, particularly for organic schemes [11]. The development of the disease depends on the relationship between the host (coffee plants), the pathogen (rust), and the environment (climate variability) [2,12].

Efficient rust control in organic crops includes alternative fungicidal compounds from natural sources (as secondary metabolites) and nutrition through soil management (as mycorrhizae), aiming to reduce or substitute synthetic pesticides without significantly affecting crop quality and production yields. A perfect example is essential oils from plants, with outstanding properties such as antioxidant, antimicrobial, and antifungal activities [13]. In the same way, biocontrol agents, such as antagonist fungi and bacteria, have been applied for CLR control [14,15]. However, despite their positive effects, using the alternatives mentioned before in agriculture remains surprisingly scarce [13,14,15].

On the other hand, biochar is a biomass-derived carbon-rich material subjected to the thermal decomposition of organic material in an oxygen-deficient environment at a high temperature called pyrolysis [16,17]. The pyrolysis process occurs in a reactor and transforms the organic material into different amounts of solid, liquid, and volatile products. The solid fraction, composed of fixed carbon and inorganic materials, is referred to as biochar [18,19].

Biochar can have different applications, and it has been utilized in resolving many environmental issues, such as adsorbing pollutants [20], reducing greenhouse emission gas [21], wastewater treatment [22], energy production [23], and soil remediation [24]. In addition, incorporating biochar into the soil produces beneficial effects such as carbon sequestration and soil improvement. Depending on the intrinsic characteristic of each biochar, it is possible to change a wide variety of soil properties, such as soil pH, water holding capacity, nutrient availability, bulk density, and soil aggregation [25,26,27]. Studies of biochar impacts on soil health and crop productivity have shown varied responses across soil types and management systems as it will vary between biochar types, soil types, and target species. Biochar application rates from 0.5 to 135 tonnes per hectare (t ha^−1^) have produced plant growth responses ranging from 29% to 324% [28].

However, recently biochar attracted particular interest for its ability to increase the microbial community composition and the enzymatic activities in the soil. These changes are essential for the biogeochemical effects of biochar in nutrient cycling, suppression of plant pathogens, and enhanced crop growth [29,30]. Plant pathogens are one of the significant threats to agriculture, and today, they face up using chemical products that can affect the quality of the final product and increase their toxicity level. The utilization of biochar in the soil seems that have suppression actions on the pathogen formations.

For example, Soilborne diseases caused by species in the genus *Fusarium* (one of the essential plant-pathogenic fungi) affect various crops in various climatic zones [31]. In addition, hardwood biochar increased populations of antagonists such as Pseudomonas or arbuscular mycorrhizal fungi (AMFs) [32].

Lately, Bonanomi et al. (2015) [33] reviewed and summarized the data from 13 path systems that tested the effect of biochar on plant disease. Their analysis reported that 85% of the studies showed a positive influence of biochar in reducing plant disease severity, 12% had no effect, and only 3% showed that biochar additions were conducive to plant disease.

Given the far-reaching importance of CLR, there is an urgent need for an adequate control system. Therefore, this review aims to synthesize the available knowledge on natural alternatives derived from extracts and essential oils from plants, fungi, bacteria, algae, and animals. Also, we want to report the influence of biochar in reducing plant diseases, such as rust in different plants, and how it could benefit coffee production and the health of coffee by suppressing the formation of rust fungi pathogens on the coffee plant.

These alternatives have been evaluated in vitro and in vivo as an initial step to developing promising HV control strategies suitable mainly for shaded production systems.

## 2. Defense Mechanisms of High Mountain Coffee and the Transition for Sustainable Production

In shaded schemes, the production is organic, and no chemicals are allowed. The specie of choice for this system is *Coffea arabica* (also known as high mountain coffee) because of its innate ability to adapt to heights (from 600 to 3100 m above sea level) and its outstanding flavor. In Mexico, practically all coffee is grown under shade (75% of the national production) in agroforestry systems where agroecological management predominates [5]. These systems offer significant environmental benefits such as carbon sequestration, soil rigidity, and soils rich in organic matter, and it has also been shown that these systems retard the proliferation of CLR.

However, one of the significant disadvantages of high-mountain coffee varieties is their increased susceptibility to CLR. Years ago, they were not infected due to the height of the crops [6]. There were sporadic outbreaks, but these were barely controlled thanks to temperature and relative humidity patterns. Nonetheless, these patterns have changed with global warming, and since 2013 they have severely affected mountain coffee crops. Therefore, despite the ecological benefits, this crisis has forced farmers to migrate to traditional monocultures, risking at least 60 native coffee species [2]. Fortunately, there is still hoped to solve this problem since a wide variety of biological control alternatives can be implemented for farmers. However, it is vital to understand the relationship between HV and high mountain coffee to establish the mode of action of these eco-friendlier solutions.

### 2.1. Virulence Factors of Hemileia vastatrix and Defense Mechanisms of High Mountain Coffee Plants

HV is an obligate parasite able to infect all the cultivated species in the *Coffea* genus affecting each species differently and needing no other hosts [34]. The infection cycle of HV (Figure 1) begins with the deposition of urediniospores (adhesion to the host surface can be either in the beam or underside) and only germinates during a leaf wetness period [35,36] where the percentage of humidity is above 80%, either in rain season or with heavy dew. In the dry periods, they can survive up to 6 weeks in the host leaf in a latent state [37]. The urediniospores form the asexual cycle, and it is still uncertain how they recognize their host. However, specific molecules in the leaf surface, such as oligosaccharides and polysaccharides, proteins (esterases), or glycoproteins, may serve as recognition factors [38,39]. The infection cycle is continuously repeated, and the clusters of uredia are spread by wind, rain, and occasionally insects [2,9,40]. In the *Coffea* genus, the principal mechanism of defense against CLR is monogenic resistance, based on resistance genes retrieved by evolution. This method has been widely explored to create resistant varieties against CLR, such as the timor hybrid, the first resistant coffee variety discovered [2].

According to the gene-for-gene theory (Figure 2) implemented by Flor et al., 1954 [41], a plant is capable of avoiding infection through the presence of matching pairs of juxtaposed dominant genes present both in the host (R) and the infectious agent (A) with character, meaning that they can be expressed even in heterozygous genotypes [41,42]. The genes in the host are known as resistance genes, while those in HV are named avirulence factors (genes corresponding to *Hemileia vastatrix*). Through this interaction, HV is deprived of its virulence, and the host can neutralize and eliminate the infectious agent without developing symptoms [43]. 

AA and RR represent the homozygous dominant genotype of *Coffea arabica* and HV for the CLR disease, while Aa and Rr are the dominant heterozygous genotypes. *Coffea* sp. all possess monogenic resistance (also known as race-specific or vertical resistance) against CLR; however, the resistance grade, which can be quantified, varies depending on the expression of one or more genes [41]. At times CLR resistance is not always complete due to the total adaptive capacity of HV. The mutation of the fungal avirulence genes is increasingly common. During mutation, the avirulence gene is transformed into a virulence gene, and the resistance host becomes susceptible to the disease by activating a hypersensitive response as its mechanism of first defense (Figure 3). This phenomenon is known as incomplete resistance [34].

HV has co-evolved to modify the metabolism of vulnerable phenotypes according to their needs. This mechanism involves the release of effector proteins that can inhibit the host’s immune system [44]. In the early stages of CLR, plants change due to a lack of sugars and peptides (for example, a decrease in hydrolases and oxidases). Therefore, alternative mechanisms are activated to control the infection progression, such as the increment of defense-like proteins to control the severity of HV. Some examples include phenylalanine ammonia lyases, peroxidases, superoxide dismutase chitinases, and β-1,3-glucanases [45,46]. However, in some cases, despite the effort of *Coffea* sp. to control the infection, these defense mechanisms occur too late to prevent fungal growth and sporulation effectively [47,48].

The virulence gene v5 comes from HV race II, the most aggressive fungal phenotype. A susceptible host (A) results from a compatible plant-pathogen interaction between the virulence gene from HV (v5) and the resistance gene from *C. arabica* (SH5). As a result, pathogen elicitors activate receptors on the cell membrane of *C. arabica*, leading to the activation of a hypersensitive response. In contrast, complete resistance (B) occurs when there is a genetic incompatibility between the HV virulence gene (v1) and the resistance gene from *C. arabica* (SH5). Hence, the plant-pathogen interaction is hindered, and the infection does not proliferate [2,9,46].

### 2.2. New Directions for Coffee Production: Relevance of Empowering Shaded Crops

Today’s most widely used solution is the implementation of monocultures of resistant varieties [47]. CLR-resistant varieties have been found in native species, mainly from Kenya. These have mainly come from crossing *C. arabica* with *C. canephora* and *C. liberica* [48]. Since implementing these varieties, breeders have had the challenge of obtaining resistant varieties without losing cup quality and production traits, which give *C. arabica* varieties their high commercial value [47,48,49]. However, this challenge has become minuscule compared to the loss resistance due to current climate conditions. With the accelerated proliferation of CLR, it is now necessary to deploy these resistance genes so that new races of HV do not immediately overcome them. So far, more than 40 different races of HV have been identified, with some new ones able to attack previously resistant hybrids. Nonetheless, new rust races continue to appear [50,51]. To reduce the selection rate of virulent races, the breeders of CENICAFÉ, a national coffee research center in Colombia, have created a hybrid cultivar with uniform agronomic characteristics and coffee quality but with a mixture of genes for HV resistance [47,48,52].

Additionally, the two main disadvantages derived from monocultures are the dramatic loss of genetic diversity among the wild *Coffea* species and the use of chemical fungicides [53]. In the case of the first problem, there is very little genetic diversity in coffee outside the tropical forests in southwestern Ethiopia, where *Coffea* evolved. Because of human activity (logging and expanded cultivation mainly), these forests have been reduced to less than one-tenth of their original size. Ethiopia’s Institute of Biodiversity Conservation and Research is struggling to hang on to what is left, and the Ethiopian government has prohibited the export of coffee plants and coffee beans from the country [54]. 

In the case of the second problem, the most effective fungicides are copper-containing due to their “tonic effect” on coffee plants that increase plant production while effectively controlling the fungus. Despite these advantages, it has to be applied rigorously before the plants become infected (as a kind of preventive protector), which increases its costs. In addition, copper accumulates in the soil, so prolonged use presents cytotoxic risks for plants and organisms in the ecosystem [11,47]. On the other hand, a study has shown that CLR-resistant monocultures may be more susceptible to American leaf spot disease, known in Latin America as “ojo de gallo,” caused by the fungal agent *Mycena citricolor*, and may lead to an outbreak of this fungal disease [55].

Even though it is the most successful solution, the reality is that it is only a partial solution, valid for at least another ten years. Furthermore, let us remember that there are no pests in nature, and organisms such as rust, which depends solely on coffee, will evolve faster to continue subsisting [51]. For this reason, if the objective is a practical, lasting, and ecological solution, it is necessary to move to a scheme where co-existence is sought instead of eradicating the fungus. Therefore, agroforestry systems are a perdurable solution for HV control [56].

For example, it has been proven that shade trees have a complex impact on American leaf spot disease and CLR, and multiple factors, including nutrients, shade level, microclimate, and possible ecological interactions, can have a combined effect [37,57]. Specifically, the CATIE center in Turrialba, Costa Rica, has developed a network of long-term trials from 1990 to 2020 in agroforestry systems with coffee [58,59]. In these studies, they found that the production systems with the *Caturra variety* (resistant cultivar) in full sun and with two drastic annual pruning of shade trees (low biomass contribution and high light input) with moderately organic and non-organic management, even with high levels of productivity harmed biophysical variables and environmental services. In contrast, particularly in agroforestry systems, the presence of the shade tree *Erythrina poepiggiana* improved product performance and environmental services. The systems that stood out in productivity and environmental services were *Erythrina poepiggiana*, in intensive and moderate conventional organic management. Shade-only systems of *Erythrina poepiggiana*, and *Erythrina + Chloroleucon eurycyclum* (timber tree and nitrogen fixer), both under organic management, have presented good profitability, with low costs and good valuation in environmental services [58].

Another study focused on water loss, and temperature found that the water loss was higher in the unshaded area (338 L ha^−1^) compared with the shaded system (150 L ha^−1^). Also, soil temperature was lower under shaded conditions, and there was water absorption complementarity between coffee and trees in a shaded area. Therefore, the shaded agroforestry coffee systems improve microclimate conditions and deep-water drainage compared with unshaded coffee systems [60].

Finally, a study in San Miguel Amatlán de Los Reyes, Veracruz, Mexico, showed that it is also possible to obtain a diverse income system in agroforestry systems that complement coffee production. By maintaining endemic trees, added to timber trees in the region (*Cedrela odorata*, *Robinsonella mirandae y Mastichodendron capirii*), the cost-benefit ratio obtained for the sale of forest and agricultural products indicates a more significant economic gain for the rustic coffee system ($20,784.00 per year/ha) compared to the traditional coffee system ($19,236.00 per year/ha) [59].

At first, it might seem that crops in full sun offer the optimal edaphoclimatic conditions for coffee growing and higher production than agroforestry crops. Nevertheless, this truth will only hold for a decade or two. Later, environmental degradation, primarily via soil erosion and pesticide residues, can seriously reduce productivity and environmental quality. In addition, in a scenario of accelerated global climate change (characterized by the reduction of water availability and the increase in temperature), we must not ignore that the coffee economy requires greater exploration and search for alternatives for the long term, capable of producing both in regions with optimal conditions and marginal environments [55,57,59].

## 3. Botanical Bioactive Compounds for CLR Control Suitable for Shaded Production Systems 

Biological control agents are better than chemical insecticides as they not only have the potential to increase the immune response of coffee crops but also enhance nutrition, preventing significant losses in the production and maturity of the coffee bean (Figure 3) [61,62]. Over the years, plants’ biological and chemical defense functions have been widely researched. Plant extracts are an alternative to chemical treatments and exist as a great variety of secondary metabolites such as alkaloids, flavonoids, terpenoids, and tannins. In addition, some plants possess biological activity against fungi, and many have shown promising results against HV [63,64]. Table 1 summarizes investigations using plant extracts to induce antifungal activity in coffee plant varieties.

Essential oils play a role in improving plant health. They also possess the capacity to prevent the growth of some types of fungi, occasionally even eliminating them. The inhibition of HV with oils has been studied by methods such as methanolic extraction in Soxhlet at different times, aqueous extraction employing hydro distillation, and ultrasound-assisted extraction with methanol and water as solvents. In addition, oils pose as an effective rust control agent of natural origin [73].

Research on the inhibition of pathogens with the aid of extracts is still in the early stages, with most studies reporting only the concentration in which an inhibitory effect. Thus, it is crucial to investigate the effect of plant extracts on HV more in-depth, as well as their toxicological, environmental, and economic effects. In addition, extracts can reduce the production costs of coffee farming since they biodegrade rapidly, do not pollute the environment, and are created inexpensively [74,75].

Bioactive compounds in plants are secondary plant metabolites eliciting pharmacological or toxicological effects in people and animals. Secondary metabolites are plant growth and development substances created outside their principal biosynthetic and metabolic routes [76]. They are considered products of biochemical sidetracks in plant cells and are not required for the plant’s everyday functioning. Several have been discovered to serve various critical purposes in living plants, including protection, attraction, and signaling. Most plant species appear capable of synthesizing these chemicals [77]. The presence of antifungal metabolites from bioactive compounds suggests they might interrupt the first communications between plant and uredospore [66].

Most botanicals are biodegradable and suitable for biocontrol. They are also quite gentle on natural adversaries. Some botanicals have a broad-spectrum effect, with fungicidal and insecticidal qualities in some cases. Secondary metabolites are a more environmentally friendly alternative to control CLR (Table 2) since they have shown antifungal and pesticide activity. For example, a study reported that tannins from *Moringa oleifera* could inhibit fungi cell formation. These molecules disrupt the cell membrane, emptying their internal content, destroying the fungus’s reproductive structure, denaturing their enzymes, and preventing the correct bundling of their substrates. Thus, leading to the death of the microorganism [78].

Flavonoids are compounds found in several parts fruits, vegetables and. They belong to a class of secondary metabolites in plants having a polyphenolic structure. The structures of flavonoids have antifungal, antiviral, and antibacterial activity. For example, the presence of flavonoids can identify *Coffea arabica* plants resistant to CLR; the higher the flavonoid content in leaves, the lower the intensity of leaf rust caused by HV [79]. Similarly, the essential oils retrieved from *Cymbopogon* sp., *Thymus* sp., and *Cynamomum* sp. contain large amounts of monoterpenes. Some examples are D-limonene, cineole, β-myrcene, anethole, p-anisaldehyde, carvacrol, carvone, limonene, felandrene, and pinene, among others, which are responsible for inhibiting the germination of several fungal pathogens, including HV uredospores. When in direct contact with HV uredospores, these compounds alter the permeability of cell membranes, causing the leakage of their constituents and inhibiting their reproductive capacity [42].

The use of plants as an alternative for rust control has been previously investigated. However, most experiments were conducted at the laboratory level, not in field conditions, where many varying conditions can cause different results. Therefore, there are no concrete or very effective results; in vitro tests with isolated compounds are changing and becoming more complex. More experiments with extracts seek to convert them into biological agents to control or prevent HV. Some studies compare oils that control with those that prevent coffee rust. For example, the essential oils of cinnamon, citronella, lemongrass, clove, eucalyptus, tea tree, thyme, and neem reduce the germination of HV uredospores; while the oils of thyme, clove, and citronella are the most promising for controlling the disease [80].

**Table 2 plants-11-02745-t002:** Experimental assays based on phytochemical compounds from plants with activity against CLR.

Plant	Class/Compounds	Efficacy of the Assay on Spore Germination Inhibition	Reference
*Baccharis glutinosa*	Flavonoids: Multijuginol, (*Z*)-3-hydroxy-1-(2-hydroxyphenyl)-3-phenyl prop-2-en-1-one, 3′-Methoxyquercetin and 12aβ-hydroxydeguelin.	Leaves treated with MEBs ^1^ significantly decreased the germination percentage of uredospores up to <5% as the dose increased (*p* < 0.05).	[66]
*Camellia sinesis*	Monoterpenes: Limonene, linalool, geraniol and Sesquerpitene: β-caryophyllene.	A significant reduction of severity was observed in the treatments with *C. sinensis*; they provide a fungicidal effect and growth suppressor of the causal agents.	[63]
*Bassica nigra- Piper nigerium*	Alkaloid: Piperine. Monoterpenes: sabinene, limonene, and β-pinene. Sesquerpitenes: β-caryophyllene, α-selinene, and germacrene.	No significant reduction of severity was observed in the treatments, therefore is recommended just as a preventive alternative.	[63]
*Cymbopogon* sp., *Thymus* sp. and *Cynamomum* sp.	Monoterpenes: D-limonene, cineole, β-myrcene, anethole, p-anisaldehyde, carvacrol, carvone, limonene, felandrene, pinene.	All the essential oils inhibited the germination of urediniospores at increasing concentrations.	[42]
*Cymbopogon citratus*, *Aloe barbadensis*, *Moringa oleifera*, *Nicotiana tabacum*	Monoterpeno: Citral (*C. citratus*) Anthranonic glycoside: Aloin (*A. barbadensis*) Tannins (*M. oleifera*) Alkaloid: Nicotine (*N. tabacum*)	The plant extracts are effective in inhibiting fungal spore germination. Extracts from *M. oleifera* and *C. citratus* proved to be the most effective, compared with *A. barbadensis*	[68]
*Piper aduncum* L.	Monoterpene: Piperitone	It can reduce uredospore mycelium germination in laboratory conditions.	[69]
*Ardisia compressa*	NC ^2^	Significant inhibition of the uredospore germination in vitro	[70]
*Eriobotrya japonica* *Ardisia compressa*, and *Ocimun basilicum*	Alkaloids, flavonoids, coumarins, and terpenes.	The aqueous extracts from the plants reduced the inhibition of the germination of uredospore at 0.12, 037, and 0.38 %, respectively	[70]

^1^ MEBs: Methanolic extracts of *Baccharis*; ^2^ NC: Mixture of compounds not characterized.

It has also been reported that *Nicotiana tabacum* managed to control CLR spore germination in vitro. Copper oxychloride 85% wettable powder and pure tobacco extract did not differ significantly in antifungal effects, suggesting that tobacco has the potential to control CLR. Only 100% tobacco extract was comparable to copper oxychloride, with all other concentrations weaker when compared to the standard fungicide. The inhibitory effect may be due to the alkaloid (nicotine) compound, insecticidal and antifungal. The results indicated that increasing the active ingredient increases spore germination inhibition [68]. The use of plants as an alternative for rust control has been previously investigated. However, the experiments were conducted in a laboratory, not in field conditions. The use of plant extracts as a method of controlling HV can be of significant advantage [65].

## 4. Novel Approaches: Use of Organisms and Biochar for the Management of *Hemileia vastatrix*

### 4.1. Strategies Based on Animals Implementation

Due to the current agriculture trends, using natural, organic-friendly, and low-cost alternatives is essential to reduce the incidence of HV and the foliar damage caused to coffee plants. One option that could aid in resolving this issue is using natural products derived from animal origin [81].

The first case of gastropods feeding on CLR [82] was observed on a widely distributed invasive snail described as a herbivore, apparently shifting its diet to consume the CLR. The detailed experiment observations of brightly orange-colored snail excrement on the coffee leaves’ undersurface led to the insight that there may be a snail consuming CLR spores. Therefore, both *Bradybaena similaris* and *Bulimulus guadalupensis* were analyzed to explore which snails were consuming HV. Both species were collected along with leaves containing CLR, and experiments showed that after 24 h, *B. similaris* cleared the coffee leaves of CLR spores while B. guadalupensis failed to consume any CLR [82] (Table 3).

On the other hand, *Mycodiplosis larvae* are predators of HV, with their larvae being examined by molecular techniques. Studies used the Kruskal-Wallis test, a non-parametric method that allows testing if a group of data comes from the same population. Sampling indicated a positive association between rust severity and the number of *Mycodiplosis larvae*, which is an alternative to fungicides [83]. According to Hernández et al., 2019 the roundworm extract (ExLom^®^) is a possible replacement for chemical antifungal agents. It consists of an aqueous solution known to reduce plant damage caused by some fungal diseases. In specific against HV, ExLom^®^ reduces the germination of spurs or leaf damage by rust, which can be produced through the use of coffee pulp to generate veneer composites with *Eisenia foetida*, *Eisenia andrei*, and *Perionyx excavates* [81]. The cultivation of the roundworms is rim-composed in 42 days, which can be subjected to water extraction that requires a day of incubation without the need for airing. The generated ExLom-P^®^ can be separated employing decantation. According to previous observations, it decreases the germination of royal spurs while reducing leaf damage if applied to the plant’s foliage. In a comparative study of coffee cultivars, treatment consisted of three concentrations and control of cow urine (0, 10, 20, 30%). Four foliar sprays were performed at 30-day intervals; eight incidence evaluations were performed at a 15-days interval. Unfortunately, there was no significant interaction by treatment; the disease incidence was similar in sprayed and non-sprayed plants. However, the crops are better tolerant of the disease; therefore, more investigation on the nourishing attributes of the ExLom-P^®^ is required [84].

**Table 3 plants-11-02745-t003:** Alternative use of animals for coffee rust control.

Animal	Use	Experimentation Details	Reference
*Mycodiplosis* larvae	Predator	There is a positive correlation between the severity caused by the rust of coffee (HV) and the number of *Mycodiplosis* spp. larvae.	[83]
*Bradybaena similaris* and *Bulimulus guadalupensis*	Predator	Experiments showed that after 24 h *B. similaris* cleared the coffee leaves of CLR spores while *B. guadalupensis* failed to consume any CLR uredospores.	[82]
Cattle	Urine	Treatments consisting of three concentrations of cow urine (10, 20, and 30%) reduced the incidence of CLR; however, they decreased the number of leaf injuries and enriched coffee crops.	[84]
ExLom-P^® 1^	Extract	It was found that the application of crude ExLom-P^®^ suppressed rust spore germination (0% germinated spores) on coffee leaf discs. Furthermore, they suppress diseases in leaves due to the microbial richness and the abundance of chitinase enzymes and β 1,3 glucanases. In addition, they provide promoters of metabolic defense against fungi, such as abscisic acid, jasmonic acid, and salicylic acid.	[81]
ExLom-PCJ^® 2^	Extract	It diminished the leaf damage of the coffee rust due to its microbial richness and the abundance of chitinase enzymes and β 1,3 glucanases. Therefore, the authors recommended it as a nutrition additive to increase coffee rust tolerance.	[81]

^1^ Variant of the patent ExLom^®^, an aqueous vermicompost extract for foliar application, where worms are fed on coffee pulp. ^2^ Variant of the patent ExLom^®^, which uses crab shell powder as a source of chitin.

### 4.2. Antifungal Activity and Coffee-Crop Nutrition Properties of Bacteria and Fungi

In nature, some relationships become critical factors for the prosperity of the species, such as the human body and the trillions of microorganisms living in synergy with us. Nevertheless, this relationship is not exclusive to the animal kingdom; there is a complex entity called the plant holobiont, which is made up of the plant itself and the diversity of macro and microscopic organisms that live with and within it (including fungi, yeasts, algae, and mainly bacteria) [76,85].

Focusing on bacteria exists a wide range of bacterial groups that form beneficial associations with plants; these associations can contribute to nitrogen fixation, solubilization of nutrients such as phosphorus, and protection against pathogens. They are located in the whole plant, externally and internally, in the roots, stems, fruits, or/and leaves (plant holobiont) [86]. This phytomicrobiome plays a crucial role in the plant’s defense mechanisms by which beneficial bacteria protect plants against pathogens. However, to protect their host, they must be in the right place at the right time [40].

A well-known mechanism of beneficial bacteria against HV is competition for space or nutrients since many bacteria live both within (endophytic) and on (epiphytic) tissues of plants, and their presence prevents pathogens from germinating and developing. Secondly, producing antimicrobial metabolites, such as hydrolytic enzymes, can attack the cell walls of competing fungi. The third way is by inducing systemic resistance in the plant [87,88,89] through hydrolytic compounds that degrade polysaccharides (e.g., chitin) that make up the cell walls of fungi [85] as well as synthesizing lipopeptides (e.g., iturin A, surfactins) that are antagonist against viruses, mycoplasmas, other bacteria, yeasts, fungi, and nematodes [90]. To sum up, the plant holobiont boots the immune response of coffee crops, thereby increasing its resistance to colonization by HV (Table 4) [91,92]. An example of this mechanism is the induction of Induced Systemic Resistance (ISR) caused by phengicins and surfactins produced by various Bacillus strains [86,88,93].

Brazil is the major coffee producer in the world, and since the arrival of coffee, in 1727, they have concentrated on thoroughly studying the interactions between coffee crops and their pests. Since the proliferation of CLR, natural alternatives such as HV antagonists and predators have been sought, whereas bacteria and fungi have stood out. As a result, studies on the biological control of HV have been developed. Likewise, some products are already patented for their use. A perfect example is the study of natural antagonists isolated from soil under organic schemes [98]. For this study, 393 isolates (fungi and bacteria) were obtained from the rhizosphere of organic crops. Even though only 17 presented a reduction in the infection occurrence, they also reduced the number of HV urediniospores produced by 70% per leaf [86,91].

Several authors have investigated the use of biological agents for CLR control. Haddad et al. evaluated seven bacterial isolates, copper hydroxide, and calcium silicate in organic coffee plants located in Minas Gerais, Brazil. The isolate B157 from *Bacillus* sp. reduced the intensity of rust and was as effective as copper hydroxide. In Brazil, a commercially available product called Biobac^®^ is composed of *Bacillus subtilis* 1336 [97].

On the other hand, various bacterial genera that are endophytes of coffee plants can penetrate different plant tissues and spread systematically by actively colonizing the apoplast, vassal ducts, and occasionally intracellular spaces [38,76]. The disease’s suppression occurs by antibiosis and competition for nutrients, in addition to inducing a resistance response in the plant. Some of these bacteria are *Bacillus lentimorbus*, *Bacillus cereus*, *Clavibacter michiganensis*, *Klebsiella pneumoniae*, *Pandorae pnomenusa*, *Kocuria kristinae*, *Cedecea* sp., and *Acinetobacter calcoaceticus* [78,86].

In a specific study, a research group found that *Bacillus subtilis* and *Pseudomonas fluorescens* inhibited urediniospore germination and reduced disease infestation by approximately 43% and 34%, respectively [91]. The bacterial strains 64R, 137G, and 3F (*Brevibacillus choshinensis*), 14F (*Salmonella enterica*), 36F (*Pectobacterium carotovorum*), 109G (*Bacillus megaterium*), 115G (*Microbacterium testaceum*), and 116G and 119G (*Cedecea davisae*) significantly reduced the disease severity when applied either 72 or 24 h before exposing the plant to HV Table 4 specifies the mechanism of action based on uredospores damage or reduction of the lesions in the coffee leaves. Under field conditions, the efficiency of the biological control of foliar spraying with *Bacillus subtilis* in the cultivars *Icatu* and *Mundo Novo* in Brazil was assessed [94]. The microorganism reduced rust by 24% and 17% for Icatu and Mundo Novo, respectively. Similarly, it has been found that the yeast *Pichia membranifaciens*, isolated from the soil, produces carboxylic acids with specific fungicidal action against CLR. The solution containing these acids, from *Pichia membranifaciens*, slowed the progress of the disease, even in places where the initial incidence was high, and reduced HV spore viability [61].

The effectiveness of bio-fungicides based on bacteria shows promising results under controlled conditions. For example, bacteria’s success ranges from 50% to 90% (in control of the germination of pathogenic spores and lesions in leaf tissue). However, this applies as long as they are administered before exposure to HV spores (12–72 h before); if its application is later, its effect decreases considerably [11].

On the other hand, it is essential to mention that fungi have modes of action similar to bacteria, inducing systemic resistance in plants and contributing to the general vigor of the plant [87,99]. However, they can also act as mycoparasites or hyperparasites, a process by which the mycelium of the biological control agent penetrates different structures of the pathogenic fungus and partially degrades its cells through the action of lytic enzymes such as chitinases, glucanases, and proteases (Table 4) [38,76,87]. The genus of fungi most studied as a biocontrol agent is Trichoderma, which can be used as foliar (e.g., for controlling CLR) and to suppress rhizosphere pathogens. Due to the wide variety of metabolites used as biocontrol agents (such as *Trichoderma mides*, *viridines*, *anthraquinones*, *pyrones*, *statins*, *ergosterol derivatives*, and *harziolactones*, to name a few) [39]. However, at least eight other genera have antagonized against HV [89]. Under experimental conditions, the effectiveness of fungi is higher than 80%, reducing the effects of HV in foliar tissues of coffee trees and the production of urediniospores. However, despite the promising results from laboratory studies, few studies have tested the effectiveness of fungi as control agents for CLR in the field, and those tested have shown poor performance [86].

Even though using fungi as a biocontrol is highly recommended, one of their main disadvantages is that they are less able to survive in adverse environments than bacteria [98]. Additionally, they are highly inhibited and sometimes do not survive exposure to UV-B light radiation [85]. Hence, it complicates their application in crops with little or no shade (such as monocultures). In addition to sensitivity to climatic conditions, the total inoculum concentration needed for effective CLR control is highly variable, and in most cases, the concentration under experimental conditions is not optimal under natural conditions [10,100]. Those are the main challenges that need to be overcome for fungi to be a viable control agent of CLR. However, they possess unique characteristics that may be the key to developing adequate control against CLR in shaded coffee crops.

### 4.3. From Sand to Land: Macroalgae as a Nutrition Key Factor for Infected Coffee Crops 

For any cultivar, the availability of nutrients is essential to guarantee growth, efficiency, and quality [2]. Coffee crops are not an exception. Furthermore, the plant’s nutritional status is essential to CLR’s physiological functions and development. Therefore, correct nutrition indirectly exerts an effect on the life cycle of the fungus since it strengthens the plant’s immune system, reducing the rate of the spread of the fungus in coffee plants [101]. Therefore, due to their attributes, algae may play a crucial role in efficient coffee rust control.

Algae are organisms of the Protista kingdom that can be unicellular or multicellular. They are autotrophs, meaning they produce their food from inorganic matter. Their organization is simple since they do not have differentiated tissues, and their habitat is commonly water or areas with humid environments [102,103]. Man used algae as fertilizer, especially in eastern cultures—the first reference to its use as an agricultural amendment date from 2700 BC in China. In Europe, its agricultural use extends from the twelfth century [104].

In specific, seaweed has long been used as a soil additive, mainly in coastal areas where it is easy to transport fresh or partially dried to the area to be fertilized. Algae are biostimulants due to their enzymatic richness, which hydrolyzes non-soluble compounds in the soil, and regulate the plant’s pH, demineralizing, detoxifying, and desalinating it. In addition, carbonates release carbonic anhydride, forming pores in the soil so plants can develop better. Large brown algae (such as *Ascophyllum nodosum*, *Laminaria* sp., *Fucus* sp., *Macrocystis pyrifera*, *Ecklonia maxima*, *Durvillea* sp., and *Sargassum* sp.) are the type of algae most widely used as biostimulants. Nowadays, various commercial options in the market contain large brown algae [86,104,105].

In coffee crops, their extracts offer advantages such as improvement in their growth and favoring the development of root and aerial systems [102]. Other advantages are inducing natural sprouting [106], increasing the absorption of mineral elements into the soil, and increasing resistance to climatic effects (such as frost, intense heat, dryness, and more excellent resistance to attacks by pests). Algae also have shown anti-stress properties as they help to overcome the post-transplant crisis. In addition, they increased uniformity in the fruit size and enhanced fungicidal effectivity [101,102,106].

Bioactive compounds extracted from marine sources, such as algae, can be an effective way to control the spread of *Hemileia vastatrix*, although possibly not eradicate the pathogen. Due to their richness in nutritious compounds, such as complex polysaccharides, they can adapt and develop in harsh climates [104]. This ability can be transferred to weak coffee crops to reinforce their immunity and guarantee fruit production and maturation despite CLR infection. Nonetheless, more research is required.

### 4.4. Biochar Application as a Possible Solution for Rust Diseases Management on Coffee Trees

Nowadays, there are several ways to control rust diseases in different plants, such as shading the plant, nutrition, biorational, biological control, and cupric fungicides [107]. In the last decade, biochar has been widely studied and considered a valuable and efficient tool for managing plant diseases because of its fungi-toxic effect, sorption of allelopathic and phytotoxic compounds that can harm the plant, and induction of plant resistance. In addition, biochar increases activities and abundance of beneficial microorganisms, changes in soil quality as nutrient availability, and abiotic conditions [99].

Biochar is a porous and carbon-rich material produced by the thermal decomposition of organic material in an oxygen-deficient environment at a high temperature (pyrolysis process). Biochar is rich in bioavailable nutrients for plants and provides the soil with improved physical, chemical, and biological properties. Due to its porous structure, it plays an important role in providing niches for several microorganisms, which can change the proportion of bacteria and fungi and, consequently, increase the enzymatic activity of the soil [108,109,110]. Besides altering surface area, and surface functional groups by a proper engineering process the properties of biochar can be modified and enhanced for specific applications. 

The action of biochar in the soil is complex and there have been few studies on understanding the mechanisms of enhancing soil suppression and inducing systematic plant defenses. There are several hypotheses of mechanisms such as (1) the biochar acts as fertilizer to enhance crop growth, (2) increases microbial biomass, changing the bacterial community, (3) shields the plant from pathogenic bacteria due to the ability to absorb the extracellular enzyme and or organic acid produced by the soil diseases, (4) alters the compounds secreted, affecting the chemistry of root exudates, and (5) redox activities are promoted in the rhizosphere influencing soil, microbial, and plant processes (Figure 4) [111,112].

Biochar has been also investigated as a solution for controlling foliar diseases on the plant and several studies have been conducted on chilies, onions, strawberries, and tomatoes [5,8]. The different studies highlighted that each pathogenic fungi have unique modes of infection, and biochar has the ability to control fungal diseases by using different mechanisms [111]. Biochar in the soil affects gene expression along both systemic acquired resistance (SAR) and induced systemic resistance (ISR) pathways, making it effective for reducing infection by pathogens that adopt differing infection strategies, as reported by Harel et. al., 2012 and Ezra et. al., 2011 [113,114]. Biochar can also render the plant more resistant to foliar fungal pathogens by the induction of systematic defense control and interfering with arbuscular mycorrhizal fungi (AMF). The mechanisms of control of pathogens are based on the type of biochar produced and operating parameters.

The temperature and feedstock sources are the parameters that affect biochar’s chemical and physical properties [100]. Biochar produced in lower temperatures contains more aliphatic compounds, labile C, and nutrients and is considered beneficial for the management of agricultural soils. In contrast, at high pyrolysis temperatures, the biochar produced by fast pyrolysis contains more aromatic compounds and fixed C, pH, ash content, surface area, stability, and pore size [42,96]. The type of organic sources used for biochar production also affects the properties of the biochar. Therefore, an appropriate selection of raw materials is fundamental to determining the chemical (e.g., pH, nutrients) and physical properties (e.g., bulk density, porosity, surface area) of the biochar produced. However, the interaction of different soils with biochar can modify the biochar-soil system, and consequentially, it affects the interactions of soil microbiota, interfering directly or indirectly with the population of beneficial and pathogenic organisms, determining the suppression of soils [97].

The biochar deposit on the soil affects the alkaline pH, which contributes to the growth of beneficial microorganisms and increases the availability of nutrients. In addition, the porous structure provides habitat and protection for the development of the soil microbiome. In other words, biochar does not directly affect the leaves of the plant, but it controls the pathogens on the leaves, creating an ideal environment for the plant’s roots in the soil [99]. However, the nutritional effects and the efficient control of pathogens in the soil strongly depend on the quality of the biochar produced.

The possibility of modifying soil chemical and physical properties by using biochar could be an excellent strategy to control leaf rust diseases. Proper nutrient supply and pH management can help coffee trees fend off leaf rust and other diseases [28]. The high nutrient-holding capacity of biochar makes it suitable to be charged with organic waste nutrients such as compost, manure, or animal/human urine. Biochar helps to control nutrient release, providing continuous plant essential nutrients, particularly nitrogen, without the risk of groundwater contamination. Evidence suggests that biochar compost may contain antifungal properties, though this has yet to be tested against fungi that impact coffee [115].

Biochar can also reduce soil acidity, the leading cause of disease management. Low and high pH encourages the propagation of fungi such as leaf rust [115,116]. Although coffee plants prefer slightly acidic soils, some coffee-growing regions have too acidic soils or generally poor nutrient levels. Many biochars have a high pH and can have a liming effect in soils, reducing the need for off-farm inputs of lime. In addition, biochar generally develops a high cation exchange capacity (CEC) which can improve the soil’s ability to retain many nutrients [116]. However, the outcome of several studies highlighted that the percentage of biochar in the soil benefits the soil. Frenkel et al., 2017 reported that at the current stage, biochar must be considered as an additive to use at a low concentration to have the ability to improve plant performance against pathogens. The studies on different crops using different percentages of biochar derived from different organic sources show positive effects only for the low percentage of biochar (up to 3%) [117].

A successful study (B4SS project) conducted by Anaya de la Rosa et. al., 2018 [118] in Peru demonstrated that biochar from green waste diverted from landfill and chicken manure was used successfully as an additive to the soil to control rust diseases in coffee leaves. The studies report that biochar stops coffee rust leaves from forming in the land where the biochar was added (B4SS formulation). However, the fungus spots were found on the leaves where the plant was not treated with biochar.

Biochar could be a sustainable source to improve the quality and yield of coffee and reduce rust diseases on the plant. However, the utilization of biochar does not always have favorable effects on soil, and the impacts on a particular microorganism cannot be generalized (Table 5). In addition, it can be considered non-economically sustainable. Despite its long-term stability, biochar undergoes chemical, physical, and biological changes over time [107,119], which makes qualitative and quantitative predictions of any effects very difficult. Therefore, studies on the potentially detrimental effects of biochar in agricultural soil are required [120].

## 5. Conclusions and Recommendations

The number of endemic coffee varieties lost by CLR is increasing. Mainly because the vast majority is produced through organic schemes where farmers lack the knowledge of tools necessary to confront this illness; therefore, the importance of integrating rust management measures that involve cultural, biological, and chemical control should be highlighted, considering the regulations of international organic coffee certifiers. Some rust management tactics are already proven efficient, such as implementing vermicompost for crop nutrition (which increases crop tolerance to CLR). Novel approaches arise on the use of biochar, showing positive effects inhibiting coffee rust left on crops and the possibility to increase the soil pH, retain the nutrients, and change the interaction between soil-microbiota inhibiting the spread of the fungus at the leave. However, the utilization of biochar does not always have favorable effects and its impact cannot be generalized. Moreover, natural antagonistic microorganisms such as bacteria, fungi, and algae have been widely studied due to their capacities to delay HV infection and protect coffee plants. However, it is a field of study where research under current crop conditions should be encouraged. The resistance and tolerance induced by biotic and abiotic agents remain unclear. There is a gap between the defense mechanisms activated in coffee crops and the agents in charge of activating such responses.

On the other hand, despite the laboratory research, there is a wide range of alternatives in the market (such as copper-based fungicides, microorganisms, and fertilizers based on algae, microalgae, or even animal residues). Nonetheless, most do not have scientific support on the appropriate dosage to reach effectiveness. Therefore, more efforts are required to generate scientific evidence of the agents’ performance that controls HV to know their actual impact on coffee crops and their environment.

Ultimately, due to the growing concern to avoid, or at least reduce, the application of pesticides and chemical fertilizers, in favor of sustainable and environmentally friendly alternatives, the search for beneficial microorganisms and compounds derived from microbes has become one of the most popular research topics in the field of plant-microbe interactions. Likewise, the design and implementation of agroecological management practices that promote the generation of beneficial ecological interactions that contribute to the protection of coffee trees in the medium and long term are needed. For example, correlation studies and modeling of factors such as rain, humidity, shade, temperature, and wind with the presence of coffee rust could bring to light the effect of such factors on the pathogenicity of the fungus. Furthermore, from an economic perspective, an efficient control strategy can increase coffee production yield, a current challenge for smallholders worldwide. At the same time, allowing producers access to organic markets that are highly demanded in Europe and the United States can open new markets to increase the producer’s profits. As a result, coffee producers can improve their production systems and migrate to sustainable alternatives with a higher income.

## Figures and Tables

**Figure 1 plants-11-02745-f001:**
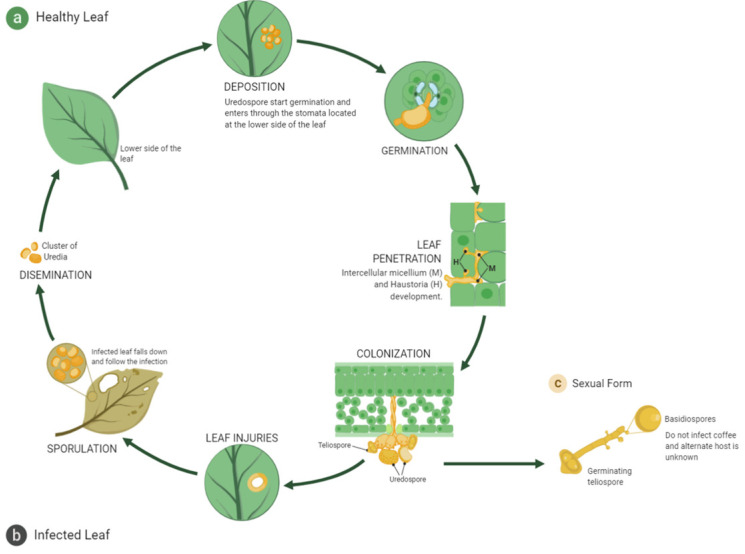
Scheme based on the infectious cycle of the fungus *Hemileia vastatrix*. Specifically, for organic cultivars (shaded production schemes), the spores finish their infection cycle in 30–60 days, and the dispersion phase begins. As the application of chemicals is limited or prohibited in organic crops, spores remain dormant, waiting for a new host to restart the infectious cycle.

**Figure 2 plants-11-02745-f002:**
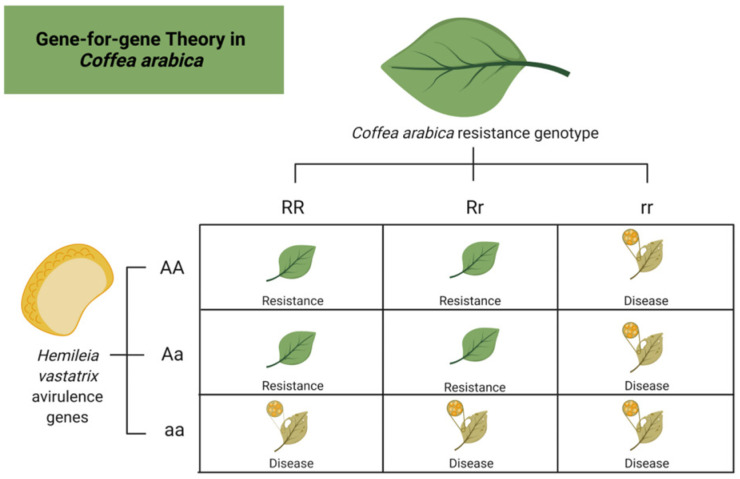
A scheme based on the gene-for-gene theory of Flor et al., 1954 [41] shows possible genotypes in the coffee leaf rust (CLR) disease between *Coffea arabica* and *Hemileia vastatrix*. AA and RR represent the homozygous dominant genotype, while Aa and Rr are the dominant heterozygous genotypes. *Coffea* sp. presents monogenic resistance (also known as race-specific or vertical resistance) against CLR, whereas the grade of resistance can be quantified, and it can vary depending on the expression of one or more genes.

**Figure 3 plants-11-02745-f003:**
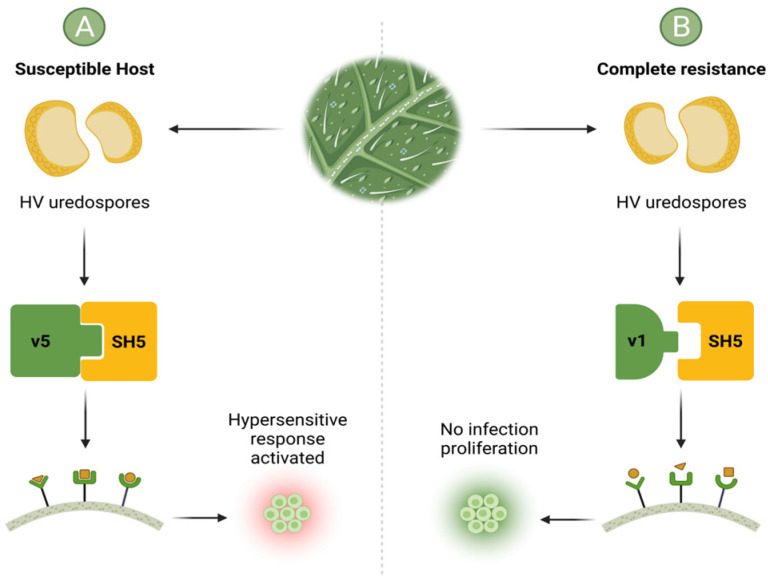
Development of complete resistance and susceptible phenotypes based on the interaction between the virulence genes from *H. vastatrix* (HV) and resistance genes from *Coffea arabica*. The virulence gene v5 comes from HV race II, the most common variety in Mexico. The susceptible host (**A**) is generated when a compatible Plant-Pathogen interaction occurs. In other words, because of the perfect match between the virulence gene from HV (v5) and the resistance gene from *C. arabica* (SH5). As a result, the pathogen elicitors activate receptors on the cell membrane in *C. arabica*, activating the hypersensitive response as a defense mechanism against coffee leaf rust (CLR). In contrast, complete resistance is present (**B**) because of the genetic incompatibility between the HV virulence gene (v1) and the resistance gene from *C. arabica* (SH5). Hence, the Plant-Pathogen interaction is not formed, and the infection does not proliferate.

**Figure 4 plants-11-02745-f004:**
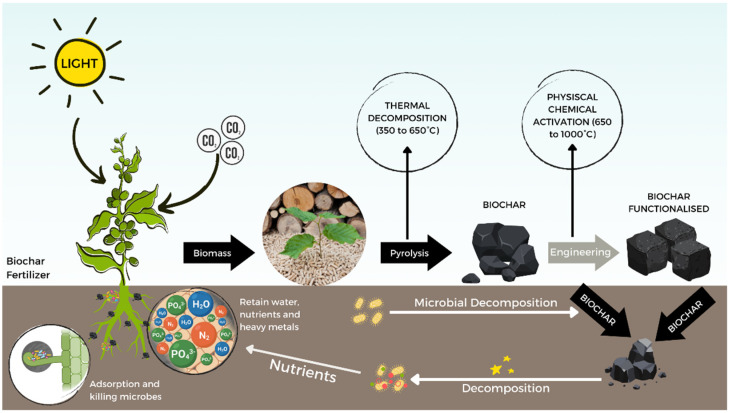
Cycle of biochar production, modification and applications in the soil and mechanism of protection from diseases.

**Table 1 plants-11-02745-t001:** Compilation of plant extracts with activity against *Hemileia vastatrix* used in different varieties of *Coffea arabica*.

*Coffea arabica* Variety	Extract Plant	Biocontrol Mechanism	Reference
Caturra	Acetone and ethanol extracts of *Ricinus communis, Datura ferox, Mansoa alliacea Tribulus terrestris*, and *Acacia farnesiana*	Inhibition of the *Hemileia vastatrix* uredospore germination	[65]
Caturra	Alcoholic extract of chilca roots (aka *Baccharis glutinosa*)	Preventive effect and reduction of foliar damage in coffee trees if it is applied 24 h before exposure.	[66]
Typica and Caturra	Extract of *Cinnamomum verum, C. sinensis, Larrea tridentata, Eucalyptus globulus, Brassica nigra*, and *Piper nigrum*	Preventive effect of reducing incidence and severity of coffee rust after the application of commercial products	[63]
*Coffea arabica* L.	Supercritical extract of *Lippia graveolens*	Antifungal effects on *Hemileia vastatrix* uredospores in vitro	[67]
Catucaií 2SL, Catuaií IAC 62 and Mundo Novo 379/19	Essential oil of cinnamon, citronella, lemongrass, clove, tea tree, thyme, neem, and eucalyptus	Inhibition of the germination of urediniospores, antimicrobial agents of the terpene group, and guaiacol, with doses equivalent to 1000 µL L^−1^	[42]
*Coffea arabica* L.	Botanical extracts of *Cymbopogon citratus, Aloe barbadensis, Moringa oleifera*, and *Nicotiana tabacum*	Inhibition of the germination of *Hemileia vastatrix* uredospores	[68]
*Coffea arabica* L.	Ethanolic extract from leaves of *Piper aduncum* L.	Uredospore mycelium germination inhibition in vitro	[69]
*Coffea arabica* L.	Aqueous extracts from leaves of *Ardisia compressa, Eriobotrya japonica, Ocimun basilicum*	Novel aqueous extract with antifungal activity	[70]
*Coffea arabica* L.	Oil of *Eremanthus erythropappus* leaves	Inhibition of the germination of *Hemileia vastatrix* uredospores	[71]
*Coffea arabica* L.	Extract from leaves of *Allium sativum* and *Vernonia polysphaera*	Inhibition of the germination of *Hemileia vastatrix* uredospores in vitro	[70]
*Coffea arabica* L.	Extracts from bulbs of *Allium sativum*, leaves of *Vernonia polysphaera*, and flower buds of * Syzygium aromaticum*	Inhibition of mycelial growth in vivo	[72]
*Coffea arabica* L.	Essential oil and extract of *Cymbopogon nardus* leaves	Inhibition of the germination of *Hemileia vastatrix* uredospores	[71]

**Table 4 plants-11-02745-t004:** Uses of bacteria and fungi against *Hemileia vastatrix*.

Specie	Inoculum Concentration	Reduction of Lesions in Coffee Leaves (%)	Reduction of Uredospores Germination (%)	Biocontrol Mechanism	References
Bacteria *Bacillus thuringiensis*	NR ^1^	76–96	NR ^1^	They induce systemic resistance in coffee trees and the production of hydrolytic enzymes (β-1,3-glucanase and chitinase) in the tissues of the leaves.	[78]
Bacteria *Bacillus lentimorbus*	1 × 10^8^ CFU ^2^	NR ^1^	50	They produce hydrolytic enzymes (β-1,3-glucanase and chitinase) and fungicidal metabolites.	[66]
Bacteria *Bacillus cereus*	1 × 10^8^ CFU ^2^	NR ^1^	50	They produce hydrolytic enzymes (β-1,3-glucanase and chitinase) and fungicidal metabolites.	[85]
Bacteria *Bacillus subtilis*	From 1 to 4.3 × 10^8^ CFU ^2^	87	100	Natural antagonist, Induces systemic resistance in coffee trees, and production of metabolites with fungicidal activity	[86]
Bacteria *Pseudomonas fluorescens*	From 1 to 4.3 × 10^8^ CFU ^2^	36	64	Natural antagonist, Induces systemic resistance in coffee trees, and production of metabolites with fungicidal activity	[94]
Bacteria *Salmonella enterica*	1 × 10^8^ CFU ^2^	74 ^3^	NR ^1^	Induction of systemic plant resistance and colonization of infection sites.	[78,95]
Fungus *Lecanicillium* spp.	5 × 10^6^ spores	NR ^1^	68 after five days of application	Hyperparasitism	[42]
Fungus *Calcarisporium* sp.	5 × 10^6^ spores	NR ^1^	51% after five days of application	Hyperparasitism	[39]
Fungus *Simplicillium* spp.	5 × 10^6^ spores	NR ^1^	89% after one day of application	Hyperparasitism	[39]
Bacteria *Pectobacterium carotovorum*	1 × 10^8^ CFU ^2^	55 ^3^	NR ^1^	Bacterias induce systemic plant resistance. They also colonized the infection sites.	[39]
Bacteria *Brevibacillus choshinensis*	1 × 10^8^ CFU ^2^	NR ^1^	9–28	They induce systemic plant resistance. They also colonized the infection sites.	[39]
Chitosan oligomers from fungal classes of *Basidiomycetes*, *Ascomycetes*, *Zygomycetes*, and *Deuteromycetes*	NS ^4^	NR ^1^	99% on coffee leaf discs	Antifungal activity, through inhibition of the germination of HV spores	[39]
Fungus *Fusarium* spp.	1 × 10^6^ spores	83–86	95–99 after 40 days of application	NR ^1^	[96]
Fungus *Penicillium* spp.	1 × 10^6^ spores	80–92	90–98 after 40 days of application	NR ^1^	[97]
Fungus *Acremonium* sp.	1 × 10^6^ spores	84	91 after 40 days of application	NR ^1^	[97]
Fungus *Cladosporium* sp.	1 × 10^6^ spores	89	96 after 40 days of application	NR ^1^	[97]
Fungus *Aspergillus* sp.	1 × 10^6^ spores	97	97 after 40 days of application	NR ^1^	[97]

^1^ NR: Not reported; ^2^ CFU: Colony Forming Units; ^3^ In the number of pustules; ^4^ NS: This measure is unsuitable.

**Table 5 plants-11-02745-t005:** Biochar impact on water, erosion and soil salinity [120].

Unfavourable Effect	Biochar Impact
Reduced availability of soil water	Reduced moisture retention and water content, negative effects on crop yields
Soil erosion	Particulate matter emissions, acceleration of biochar degradation, loss of soil fertility
Low biodegradability	Low environmental sustainability due to the accumulation on the soil for decades
Rise in soil salinity	Plant growth inhibition, negative effects on crop yields and economic impact
Excessive increase in soil pH	Inhibited plant growth due to precipitation and availability of nutrients, extreme pH, changed mobility of PTEs.
Excessive sorption of nutrients	Nutrient immobilisation and reduced bioavailability for plants and microflora, plant growth inhibition, reduced yields
Formation of toxic PAHs	Toxicity to soil macro- and microbiota, increased human health risks in case of PAHs distribution in environment and their accumulation in the crop biomass/food chain
Formation of toxic VOCs	Plant growth inhibition, human health risk in case of VOCs distribution in environment and their accumulation in the crop biomass
Presence of PTEs	Decreased plant growth, inhibition, mortality, genotoxic effects, human health risk in case of PTEs distribution in environment and their accumulation in the crop biomass/food chain
Formation of toxic dioxins	Human health risks in case of dioxin distribution in the environment and their accumulation in the crop biomass/food chain
Changes in microbial communities	Shifts in the fungi-to-bacteria ratio, decreased microbial activity, N mineralisation, SOC sequestration
Adverse effects of biochar on soil invertebrates	Reproduction and growth inhibition, mortality, genotoxicity—decrease in biochar incorporation, soil enzyme activity and thus plant productivity

## Data Availability

Not applicable.

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
