# Peer review of "Towards an Eco-Friendly Coffee Rust Control: Compilation of Natural Alternatives from a Nutritional and Antifungal Perspective"

_plants, 2022, doi:10.3390/plants11202745_

Round 1

Reviewer 1 Report

The article presents a review of the use of coffee leaf rust control strategies and tactics. According to the review, the authors point to the use of a product called biochar for the control of coffee leaf rust. However, there is no detailed information about this product, as well as the mechanism of action on plants and fungus. In addition, the product is presented as a solution without disadvantages of use and this is a concern of the critical scientist, who considers that the use of a management strategy or tactic can have both advantages and disadvantages. Thus, we recommend that the authors present more critical arguments about the use of these products, including advantages and disadvantages of use under different soil conditions and genetic diversity of coffee plants. The authors seem to be selective in citing articles about coffee rust, and this is not a very scientific practice either. After these modifications the manuscript can be accepted.

Author Response

Dear reviewer,

 We wish to thank your thoughtful feedback on this paper. We have addressed your suggestions as follows in the paragraphs below:

The article presents a review of the use of coffee leaf rust control strategies and tactics. According to the review, the authors point to the use of a product called biochar for the control of coffee leaf rust. However, there is no detailed information about this product, as well as the mechanism of action on plants and fungus. In addition, the product is presented as a solution without disadvantages of use and this is a concern of the critical scientist, who considers that the use of a management strategy or tactic can have both advantages and disadvantages. Thus, we recommend that the authors present more critical arguments about the use of these products, including advantages and disadvantages of use under different soil conditions and genetic diversity of coffee plants. The authors seem to be selective in citing articles about coffee rust, and this is not a very scientific practice either. After these modifications, the manuscript can be accepted.

Author´s response: Thank you for the assertive suggestion. Additional information regarding biochar has been added to the main text (Please see lines 554-562, 564-571 and 579-589) Analysing the literature, there are not yet studies that report the specific mechanisms of biochar on the rust HV infection, but there is a large literature that reports the fungicide action of the biochar in general on plant pathogens. The general effect and mechanism of the biochar on pathogens have been added to the review (Please see lines 564-571 and 579-589), and Figure 4 was incorporated to explain the cycle of the biochar and the mechanism of protection from diseases.

The disadvantage of biochar, such as the high production cost and the unknown mechanisms of action, was discussed (Please see lines 642-649). Also, we constructed Table 5 to inform the other side of the unfavorable effects.

The selective in citing articles about coffee rust is due to the limited studies carried out on this specific coffee disease. Hence this review has the scope to encourage the scientific community to investigate the mechanisms of biochar and expand their studies related to this important problem and find a solution that prevents the issues caused by using chemical fungicides.

Reviewer 2 Report

In the study entitled "Towards an eco-friendly coffee rust control: Compilation of natural alternatives from a nutritional and antifungal perspective. " by Torres et al., the authors made a revisión of some approaches for CLR control.
I find the review of interest in the field given the importance of coffee in the world economy. Besides, many families' economy depends as small producers on this activity.  
The review gives us a soon approach to organic developing strategies for CLR control. The reader of this review can have a starting point for further research.
I found some inconsistencies in the manuscript; some of them are:
Line 51; clarified 14 of 32 states.
Line 59; clarified "where" families abandon their properties (in Mexico?)
Line 90; please explain why this wide range of 0.5 to 135 t ha-1. It seems nonsense.
Line 187, inconsistency in the phrase
Line 285, incomplete phrase
Line 293, a missing point
Line 332, inconsistency H.V.
Lines 421-422, seem inconsistent
Line 559, Ant antagonist?

Author Response

Dear reviewer,

We wish to thank your thoughtful, valuable feedback on this paper. We improved the manuscript accordingly to your recommendations.

In the study entitled "Towards an eco-friendly coffee rust control: Compilation of natural alternatives from a nutritional and antifungal perspective. " by Torres et al., the authors made a revisión of some approaches for CLR control.
I find the review of interest in the field given the importance of coffee in the world economy. Besides, many families' economy depends as small producers on this activity.  
The review gives us a soon approach to organic developing strategies for CLR control. The reader of this review can have a starting point for further research.

I found some inconsistencies in the manuscript; some of them are:

Author´s response: Thank you for your comments. We have addressed your suggestions as follows in the paragraphs below:

Line 51; clarified 14 of 32 states.

Author´s response: Thank you for your valuable comment, the sentence was clarified as follow: 14 states of the country. Please see line 53.

Line 59; clarified "where" families abandon their properties (in Mexico?)

Author´s response: Thank you for your assertive comment. The sentence was completed. Please see line 61.

 Line 90; please explain why this wide range of 0.5 to 135 t ha-1. It seems nonsense (Biochar).

Author´s response: Thank you for your observation. The biochar characteristics are drastically affected by the operating conditions and type of feedstocks used. The type of feedstock also affects the amount of nutrients contained in biochar. In the literature, the value of biochar that benefits the plant is reported in the range reported above. The paragraph was reconstructed as follow: “Studies of biochar impacts on soil health and crop productivity have shown varied responses across soil types and management systems as it will vary between biochar types, soil types, and target species. Biochar application rates from 0.5 to 135 tonnes per hectare (t ha-1) have produced plant growth responses ranging from 29% to 324% (Glaser et.al., 2002).

            Glaser, B.; Lehmann, J.; Zech, W. Ameliorating Physical and Chemical Properties of Highly Weathered Soils in the Tropics with Charcoal – a Review. Biol. Fertil. Soils 2002 354 2002, 35, 219–230, doi:10.1007/S00374-002-0466-4.

Please see lines 91-94 in the Manuscript.

Line 187, inconsistency in the phrase

Author´s response: Thank you for pointing this out. The phrase was reconstructed. Please see lines 152-154.

 Line 285, incomplete phrase

Author´s response: Thank you for your observation. The phrase was completed. Please see lines 332-336.

Line 293, a missing point

Author´s response: Thank you for your valuable comment. The paragraph was recontrsucted accordingly. Please see lines 341-348.

 Line 332, inconsistency H.V.

Author´s response: Thank you for pointing this out. The inconsistency was corrected. Please see line 385.

 Lines 421-422, seem inconsistent

Author´s response: Thank you for your good suggestion. The paragraph was reconstructed. Please see lines 471-475.

  Line 559, Ant antagonist?

Author´s response: Thank you for your comment. The correction was done. Please see line 664.

Reviewer 3 Report

Very interesting review paper on the potential of botanical compounds to control CLR. However, suggest to delete par. 2.1 as this information is already well documented elsewhere in several scientific papers and handbooks on coffee. Besides, host resistance is the most cost-effective way for smallholder coffee growers to control CLR (and CBD) and has been successfully applied in coffee producing countries like Colombia, Kenya and Tanzania.

Author Response

Dear reviewer,

We wish to thank your thoughtful feedback on this paper. We have addressed your suggestions as follows in the paragraph below:

Very interesting review paper on the potential of botanical compounds to control CLR. However, suggest to delete par. 2.1 as this information is already well documented elsewhere in several scientific papers and handbooks on coffee. Besides, host resistance is the most cost-effective way for smallholder coffee growers to control CLR (and CBD) and has been successfully applied in coffee producing countries like Colombia, Kenya and Tanzania.

Author's response: Thank you for your assertive suggestion. We agree with your point related to the CLR life cycle. Section 2 was reconstructed, and in the revised version, section 2.1 was shortened to explain the virulence factors and defense mechanisms briefly.  New section 2.2 was added to discuss the two sides of the use of the resistance coffee plants and the transition to more sustainable production schemes. Please see lines 217-293.

Round 2

Reviewer 1 Report

After the revisions have been made the manuscript can be accepted.